# Phosphorus-Derived Isatin Hydrazones: Synthesis, Structure, Thromboelastography, Antiplatelet, and Anticoagulation Activity Evaluation

**DOI:** 10.3390/ijms26136147

**Published:** 2025-06-26

**Authors:** Aleksandr V. Samorodov, Wang Yi, Dmitry A. Kudlay, Elena A. Smolyarchuk, Alexey B. Dobrynin, Ayrat R. Khamatgalimov, Karina Shchebneva, Marina Kadomtseva, Dilbar Komunarova, Anna G. Strelnik, Andrei V. Bogdanov

**Affiliations:** 1Department of Pharmacology, Bashkir State Medical University, Lenin St. 3, Ufa 450008, Russia; 2School of Pharmacy, Hangzhou Normal University, Yuhangtan, 2318, Hangzhou 310030, China; yi.wang1122@hznu.edu.cn; 3Department of Pharmacology, Sechenov First Moscow State Medical University (Sechenov University), Bolshaya Pirogovskaya St. 2, Building 4, Moscow 119435, Russia; kudlay_d_a@staff.sechenov.ru (D.A.K.); smolyarchuk@mail.ru (E.A.S.); 4Institute of Immunology FMBA of Russia, Kashirskoye Highway, Building 24, Moscow 115522, Russia; 5Arbuzov Institute of Organic and Physical Chemistry, FRC Kazan Scientific Center, Russian Academy of Sciences, Akad. Arbuzov St. 8, Kazan 420088, Russia; aldo@iopc.ru (A.B.D.); khamatgalimov@gmail.com (A.R.K.); shchebnevak@mail.ru (K.S.); nikanna@iopc.com (A.G.S.); 6Department for Design and Techiques of Electronic Tools Manufacturing, Institute for Radio-Electronics and Telecommunications, Kazan National Research Technical University Named After A. N. Tupolev-KAI, Karl Marx St. 10, Kazan 420111, Russia; 7A.M. Butlerov Institute of Chemistry, Kazan Federal University, Kremlevskaya St. 29, Kazan 420008, Russia; kozyreva_marina_@mail.ru; 8Department of Organic Chemistry, Kazan National Research Technological University, Sibirsky Trakt 12, Kazan 420029, Russia; dil6744@mail.ru

**Keywords:** isatin, phosphorus, hydrazones, crystal structure, hemostasis, antithrombotic activity

## Abstract

A series of new isatin hydrazones bearing phosphorus-containing moiety was synthesized through a simple, high-yield and easy work-up reaction of phosphine oxide (Phosenazide) or phosphinate (2-chloroethyl (4-(dimethylamino)phenyl)(2-hydrazinyl-2-oxoethyl)phosphinate, CAPAH) hydrazides with aryl-substituted isatins. The ^31^P NMR technique showed that, in most cases, out of 12 examples in solution, the ratio of the two spatial isomers varied from 1:1 to 1:3. Quantum chemical calculations confirmed the predominance of *Z,syn* form both in the gas phase and in solution. According to X-ray analysis data in crystals, they exist only in *Z,syn* form too. Most of the phosphine oxide derivatives and 5-methoxy- and 5-bromoaryl phosphinate analogs exhibit anti-aggregant activity at the level of acetylsalicylic acid but inhibit platelet activation processes more effectively. The 5-chloro type phosphinate derivative exhibits anti-aggregant properties more effectively than acetylsalicylic acid under the conditions of the tissue factor (TF)-activated thromboelastography (TEG) model, the ex vivo thrombosis model. Thus, all the obtained results can become the basis for future pharmaceutical developments to create effective anti-aggregation drugs with broad antithrombotic potential.

## 1. Introduction

Thrombosis and thromboembolic complications are still widespread and high in frequency, being one of the main causes of death and disability in the adult population in industrialized countries [1]. Disturbances in the hemostasis system cause the widespread occurrence of thrombosis of various origins and are a key link in the pathogenesis of many diseases and critical conditions [2]. The need to use means of pharmacological correction of the hemostatic system and the degree of their effectiveness is a fact proven by the results of more than 280 meta-analyses of secondary and six trials of primary prevention of thrombosis [3]. One of the most popular approaches to the creation of innovative drugs is the construction of hybrid molecules by decorating the basic chemical scaffold with various pharmacophoric fragments carrying a certain functional load.

Isatin is a well-known heterocyclic privileged scaffold in many clinically approved drugs used for the treatment of different diseases [4,5,6,7,8,9,10]. The focus of research into the pharmacologically important properties of isatin and its derivatives often falls on isatin-3-hydrazones that exhibit antimicrobial [11], neuroprotective [12,13], psychoactive [14], anticancer [15], and other activities (Figure 1).

Organophosphorus compounds are widely used in medicine. The discovery of new phosphorus-based drugs with various types of biological activity is a modern trend in medicinal chemistry and pharmaceuticals [16,17,18,19,20,21,22]. Among the large structural diversity of this class of compounds, phosphine oxides are of particular interest, as they exhibit antitumor, neuroleptic, fungicidal, antiviral, and other types of activity [23,24,25,26,27].

Herein, we report the synthesis of a series of new phosphorus-containing hydrazones based on aryl-substituted isatins, with the aim of evaluating their anti-aggregational and anticoagulation activities. The development of the class of phosphorus-containing isatin acyl hydrazones currently faces, in our opinion, two problems. Thus, as limitations to its expansion, it is necessary to note the extremely small number of synthesized phosphorus-containing hydrazides and their low solubility in water. However, variation in substituents in the aromatic ring, at the nitrogen atom of isatin, and the phosphorus atom will improve the physicochemical properties and bioavailability of these molecules.

## 2. Results and Discussion

### 2.1. Chemistry

#### 2.1.1. Synthesis of Phosphorus-Containing Isatin Hydrazones

The target hydrazones **3a–f** were obtained by an acid-catalyzed reaction of aryl-substituted isatins **1a–f** with hydrazide **2,** which contains a phosphine oxide moiety, yielding high results (Figure 1). Their structure and purity were unequivocally proven by IR and NMR spectroscopy, mass-spectrometry, and elemental analysis data (Appendix A).

Among the variety of types of biological activity of organophosphorus compounds, derivatives of phosphinic acid R_2_P(O)(OR) are especially worth noting [28,29,30]. In this regard, the synthesis of hybrids of isatins and hydrazide **4** (CAPAH), which exhibits neuroleptic properties [31,32], was of interest (Figure 2).

It is known that acyl hydrazones, both in solution and in the crystalline state, can exist in the form of four spatial forms [33,34]. In the case of isatin-3-acyl hydrazones, such isomerism can be represented by Figure 3.

It is reasonable to assume that compounds **3a–f, 5a–f** should be predominantly in the form of Z-isomers due to the possibility of forming an intramolecular hydrogen bond O···H. On the other hand, the solvent also influences mutual isomeric transitions [34]. Thus, the ^31^P NMR spectra in DMSO contain two signals in all cases. In the series **3a–f** and **5a–f**, the ratio of isomers varies from 1:1 to 1:1.8 and from 1:2 to 1:3, respectively (Figures of 31P NMR spectra in the Appendix A).

To explain this observation, we performed quantum chemical calculations, which revealed that the *Z*-forms of both conformers (synperiplanar and antiperiplanar) are energetically more favorable, whereas the *E*-forms exhibit slightly higher kinetic stability (Table 1). Taking into account the influence of the DMSO solvent, the calculations also confirmed the strength of the hydrogen bonds and the thermodynamic stability of the *Z*-forms. Furthermore, in DMSO, the intramolecular O···H hydrogen bond becomes stronger, with the distance decreasing from 2.03 to 2.06 Å to 2.02 Å. This finding is consistent with the experimental NMR observations.

#### 2.1.2. X-Ray Study

According to X-ray data (Figure 2A,B, Appendix A), compounds **3e** and **5e** crystallize in different monoclinic space groups: compound **3e** crystallizes in space group *C2/c*, and compound **5e** crystallizes in the racemic space group *P2_1_/c*. The asymmetric unit of the compound **5e** contains two crystallographically independent molecules with similar geometries. Among all possible isomers indicated in Figure 3, it is the *Z,syn* form that is realized in crystals of compounds **3e** and **5e**.

The conformations of the molecules in the crystals are similar: the hydrazone fragments are planar, and the nitrogen atom N1 has planar trigonal coordination as typical for amides. The rotation of the phosphorus substituent relative to the carbonyl group of the hydrazone fragment in the crystals of compounds **3e** and **5e** is different: the torsion angle O8C8C9P1 in the crystal of compound **3e** is −104.02(14)°, while in compound **5e** the torsion angle O8AC8AC9AP1A is 105.03(8)° and O8BC8BC9BP1B is −97.42(8)°. The C=O group of the oxindole fragment is directed toward the N-H bond of the hydrazone unit so that it forms an intramolecular hydrogen bond. In this case, the main geometric parameters (bond lengths and bond angles) in molecules **3e** and **5e** are usual.

Analysis of intermolecular interactions in the crystal of compound **3e** showed the presence of intramolecular N-O···H-type hydrogen bonds and multiple short C-O···H-type contacts involving all oxygen atoms present in this molecule. A short C-O···F-type contact is also present in this crystal. These contacts lead to the formation of a three-dimensional network. The same short contacts are present in the crystal of compound **5**e.

Thus, using simple and convenient synthetic procedures, we obtained various aryl-substituted phosphorus-containing isatin derivatives to establish their activity in relation to various factors of hemostasis.

### 2.2. Biological Studies

A wide range of biological activities was studied for newly synthesized phosphorus-containing isatin-3-hydrazones in order to understand the further therapeutic potential of these substances. The studies included experiments to study the effect on the hemostasis system using a global test—thromboelastography—with subsequent evaluation of the processes of activation and aggregation of platelets, the coagulation link of hemostasis. Below, we will dwell in detail on each type of property, and at the end, summarize which modification leads to the best bioeffect.

#### 2.2.1. Anticoagulant and Antiplatelet Activities

In this work, anticoagulant and antiplatelet properties were studied (Table 2).

Phosphine oxides **3a, 3d–f,** and phosphinates **5a–c** exhibit antiplatelet activity at the level of acetylsalicylic acid in terms of reducing the maximum amplitude of platelet aggregation. At the same time, these compounds are more effective than acetylsalicylic acid in prolonging the lag period. Compound **5d** exceeds the values of acetylsalicylic acid in terms of the degree of aggregation reduction (13.7 vs. 17.4 at *p* < 0.05). With respect to the coagulation link of hemostasis, these compounds showed an effect exclusively on the APTT index. It should be noted that the results of APTT elongation different from the control were recorded in compounds **3c**, **3d, 3e**, **3f**, **5a**, **5b, 5d,** and **5f**. Therefore, the resulting compounds have a high potential as a scaffold for the development of effective anticoagulant and antiplatelet agents.

#### 2.2.2. Fluorescence-Activated Cell Sorting (FACS) Analysis

The results illustrating the effects of the tested compounds on the platelet activation are shown in Table 3.

It was found that acetylsalicylic acid does not affect the expression level of CD62. All studied compounds effectively reduced the level of *p*-selectin expression to values close to those of intact platelets. Not all synthesized compounds cause platelet activation and inhibit ADP-induced CD 62 expression.

#### 2.2.3. Thromboelastography

Thromboelastography (TEG) is necessary for global and dynamic assessment of the hemostatic system as a result of the interaction of cellular proteins and plasma proteins at the stages of initiation, formation, and lysis of a blood clot. Registration of TEG, activated by excess tissue factor, is an in vitro model that simulates the hypercoagulation of patients at the time of real thrombosis, allowing for an assessment of the effectiveness of compounds in a complex manner on initially compromised platelets, active blood coagulation factors, and the fibrinolysis system [35]. The results of the study are presented in Table 4.

The predominantly anti-aggregatory activity of the synthesized compounds in TF-activated platelets was established. It was established that compound **5d** exceeds the activity indices of acetylsalicylic acid and affects the functional activity of platelets (MA) and the coagulation component of the hemostasis system (R), which leads to the fact that the forming clot is functionally incomplete (G). The effect of the compounds on the fibrinolysis system was not recorded (CLT). The remaining compounds are inferior to acetylsalicylic acid in terms of anti-aggregation activity in equimolar concentrations under these experimental conditions.

Thus, all the obtained results on the biological activity of the synthesized compounds can lay the groundwork for future pharmaceutical developments for the creation of effective antiplatelet drugs due to their wide antithrombotic potential (Figure 3).

## 3. Materials and Methods

### 3.1. Chemistry

IR spectra were recorded on an IR Fourier spectrometer Tensor 37 (Bruker Optik GmbH, Ettlingen, Germany) in the 400–3600 cm^−1^ range in KBr. The ^1^H-, ^31^P-, and ^13^C-NMR spectra were recorded on a Bruker AVANCE 400 spectrometer (Bruker BioSpin, Rheinstetten, Germany) operating at 400 MHz (for ^1^H NMR), 162 MHz (for ^31^P NMR), and 101 MHz (for ^13^C NMR), a Brucker spectrometer AVANCE*III*-500 (Bruker BioSpin, Rheinstetten, Germany) operating at 500 MHz (for ^1^H NMR), 203 MHz (for ^31^P NMR), and 126 MHz (for ^13^C MMR), and a Bruker AVANCE 600 spectrometer (Bruker BioSpin, Rheinstetten, Germany) operating at 600 MHz (for ^1^H NMR), 243 MHz (for ^31^P NMR), and 151 MHz (for ^13^C NMR). Chemical shifts were measured in δ (ppm) with reference to the solvent (δ = 2.50 ppm and 39.50 ppm for DMSO-*d*_6_, for ^1^H and ^13^C NMR, respectively) or to internal standard H_3_PO_4_ (for ^31^P NMR). Mass spectra ESI and MALDI were obtained on AmazonX (Bremen, Bruker, Germany) and UltraFlex III TOF/TOF (Bremen, Bruker, Germany) spectrometers, respectively. Elemental analysis was performed on a CHNS-O Elemental Analyser EuroEA3028-HT-OM (EuroVector S.p.A., Milan, Italy). The melting points were determined on the Stuart SMP10 apparatus (Birmingham, UK).

**X-ray crystallography data.** Data of **3e** and **5e** were collected on a Bruker D8 QUEST with PHOTON II CCD diffractometer (Bruker AXS, Germany), using graphite monochromated MoKα (λ = 0.71073 Å) radiation and ω-scan rotation. Data collection images were indexed, integrated, and scaled using the APEX3 [36] data reduction package and corrected for absorption using SADABS [37]. The structure was solved by direct methods and refined using the SHELX program [38]. All non-hydrogen atoms were refined anisotropically. H atoms were calculated on idealized positions and refined as riding atoms. Crystal Data and Refinement Details are presented in Appendix A (see Appendix A). The X-ray analysis was performed on the equipment of the Spectral Analytical Center of FRC Kazan Scientific Center of RAS.

CCDC 2467006, 2467007 (**3e** and **5e**) contain the supplementary crystallographic data for this paper. These data can be obtained free of charge via www.ccdc.cam.ac.uk/conts/retrieving.html or from the Cambridge Crystallographic Data Centre, 12 Union Road, Cambridge CB2 1EZ, UK; fax: (+44) 1223-336-033; or deposit@ccdc.cam.uk.


**Quantum chemical calculations.**


The molecular structures of four spatial forms of compounds **3** and **5** in the gas phase and DMSO solvent were fully optimized using DFT B3LYP functional [39,40] with 6-311 + G(d,p) basis. Geometry optimization was performed without symmetry constraints. To ensure the calculated structures were indeed minima, vibrational analyses were performed using the same methods. The Polarizable Continuum Model (PCM) is used as a salvation model with the molecule of interest inside a cavity in a continuous homogenous dielectric medium that represents the DMSO solvent. All calculations were performed using the GAUSSIAN’09 program [41]. The standard keywords in the Gaussian package were used in optimization processes.

**Synthesis of hydrazones 3a–f and 5a–f (general method).** To the mixture of corresponding isatin **1a–f** (10 mmol) and 15 mL of absolute ethanol, corresponding hydrazide **2** or **4** (10 mmol) and three drops of trifluoroacetic acid were successively added. The reaction solution was heated under reflux for 3 h. After spontaneously cooling to room temperature, the precipitate formed was filtered, washed with absolute ether, and dried in a vacuum. Chemical shifts in NMR spectra are given below for signals of predominant geometric isomers.

**2-(Diphenylphosphoryl)-*N’*-(5-methyl-2-oxoindolin-3-ylidene)acetohydrazide (3a).** Orange powder. Yield 80%, m.p. = 198 °C. IR spectrum, ν, cm^−1^: 1626 (C=C), 1683 (C=O), 1711 (C=O), 3059 (CH), 3432 (NH). ^31^P NMR (162 MHz, DMSO-*d*_6_) δ 24.93. ^1^H NMR (400 MHz, DMSO-*d*_6_) δ 12.57 (s, 1H, NH), 10.66 (s, 1H, NH), 7.87–7.81 (m, 4H, Ar), 7.58–7.52 (m, 7H, Ar), 7.16 (br. d, *J* = 7.9 Hz, 1H, Ar), 6.80 (d, *J* = 8.0 Hz, 1H, Ar), 4.22 (d, *J* = 14.2 Hz, 2H, CH_2_), 2.27 (s, 3H, CH_3_). ^13^C NMR (151 MHz, DMSO-*d*_6_) δ 164.6, 163.0 (d, *J* = 12.1 Hz), 141.6, 139.5, 133.1, 132.1, 132.0 (d, *J* = 162.5 Hz), 131.9, 131.7, 131.5, 130.6 (d, *J* = 9.5 Hz), 128.7 (d, *J* = 11.4 Hz), 126.5, 110.4, 37.4 (d, *J* = 60.1 Hz), 20.4. MS (ESI): *m*/*z* = 418.0 [M+H]^+^; Found: C, 66.01; H, 4.69; N, 10.01; P, 7.22. Anal. calcd. (%) for C_23_H_20_N_3_O_3_P: C, 66.18; H, 4.83; N, 10.07; P, 7.42.

**2-(Diphenylphosphoryl)-*N’*-(5-methoxy-2-oxoindolin-3-ylidene)acetohydrazide (3b).** Orange powder. Yield 78%, m.p. = 192 °C. IR spectrum, ν, cm^−1^: 1595 (C=C), 1693 (C=O), 1711 (C=O), 2964 (CH), 3396 (NH). ^31^P NMR (243 MHz, DMSO-*d*_6_) δ 25.35. ^1^H NMR (600 MHz, DMSO-*d*_6_) δ 12.58 (s, 1H, NH), 11.04 (s, 1H, NH), 7.86–7.81 (m, 4H, Ar), 7.59–7.50 (m, 7H, Ar), 6.94 (br. d, *J* = 10.0 Hz, 1H, Ar), 6.83 (d, *J* = 8.9 Hz, 1H, Ar), 4.24 (d, *J* = 14.5 Hz, 2H, CH_2_), 3.74 (s, 3H, CH_3_). ^13^C NMR (151 MHz, DMSO-*d*_6_) δ 167.8, 162.5 (d, *J* = 23.8 Hz), 137.1, 136.0, 134.5, 133.7, 132.4 (d, *J* = 169.2 Hz), 131.8, 130.6 (d, *J* = 9.6 Hz), 128.5 (d, *J* = 12.3 Hz), 120.3, 117.4, 106.3, 56.0, 34.4 (d, *J* = 61.4 Hz). MS (ESI): *m*/*z* = 434.23 [M + H]^+^; Found: C, 63.60; H, 4.49; N, 9.57; P, 7.02. Anal. calcd. (%) for C_23_H_20_N_3_O_4_P: C, 63.74; H, 4.65; N, 9.70; P, 7.15.

***N’*-(5-Bromo-2-oxoindolin-3-ylidene)-2-(diphenylphosphoryl)acetohydrazide (3c).** Orange powder. Yield 87%, m.p. = 203 °C. IR spectrum, ν, cm^−1^: 1603 (C=C), 1697 (C=O), 1714 (C=O), 2997 (CH), 3434 (NH). ^31^P NMR (243 MHz, DMSO-*d*_6_) δ 26.07. ^1^H NMR (600 MHz, DMSO-*d*_6_) δ 12.47 (s, 1H, NH), 11.29 (s, 1H, NH), 7.86–7.83 (m, 4H, Ar), 7.59–7.52 (m, 8H, Ar), 6.88 (d, *J* = 8.2 Hz, 1H, Ar), 4.26 (d, *J* = 14.2 Hz, 2H, CH_2_). ^13^C NMR (151 MHz, DMSO-*d*_6_) δ 164.3, 163.3 (d, *J* = 3.6 Hz), 138.0, 135.1, 133.1 (d, *J* = 159.6 Hz), 132.2, 130.6 (d, *J* = 8.8 Hz), 128.7 (d, *J* = 9.7 Hz), 123.2, 116.7, 113.6, 113.0, 37.4 (d, *J* = 58.8 Hz). MS (ESI): *m*/*z* = 482.10 [M + H]^+^; Found: C, 54.60; H, 3.48; Br, 16.43; N, 8.60; P, 6.29. Anal. calcd. (%) for C_22_H_17_BrN_3_O_3_P: C, 54.79; H, 3.55; Br, 16.57; N, 8.71; P, 6.42.

***N’*-(5-Chloro-2-oxoindolin-3-ylidene)-2-(diphenylphosphoryl)acetohydrazide (3d).** Yellow powder. Yield 92%, m.p. = 191 °C. IR spectrum, ν, cm^−1^: 1625 (C=C), 1697 (C=O), 2976 (CH), 3464 (NH). ^31^P NMR (243 MHz, DMSO-*d*_6_) δ 24.82. ^1^H NMR (600 MHz, DMSO-*d*_6_) δ 12.47 (s, 1H, NH), 11.31 (s, 1H, NH), 7.87–7.82 (m, 4H, Ar), 7.58–7.49 (m, 6H, Ar), 7.39 (d. d, *J* = 8.3 Hz, *J* = 1.9 Hz, 1H, Ar), 7.27 (br. s, 1H, Ar), 6.93 (d, *J* = 8.4 Hz, 1H, Ar), 4.25 (d, *J* = 14.2 Hz, 2H, CH_2_). ^13^C NMR (151 MHz, DMSO-*d*_6_) δ 168.0 (d, *J* = 5.6 Hz), 162.1, 140.9, 133.9, 133.2, 131.9, 131.4 (d, *J* = 160.2 Hz), 130.5 (d, *J* = 9.5 Hz), 128.5 (d, *J* = 11.8 Hz), 126.6, 121.2, 120.5, 112.6, 34.8 (d, *J* = 59.9 Hz). MS (ESI): *m*/*z* = 437.99 [M + H]^+^; Found: C, 60.19; H, 3.77; Cl, 8.01; N, 9.48; P, 6.87. Anal. calcd. (%) for C_22_H_17_ClN_3_O_3_P: C, 60.35; H, 3.91; Cl, 8.10; N, 9.60; P, 7.07.

**2-(Diphenylphosphoryl)-*N’*-(5-fluoro-2-oxoindolin-3-ylidene)acetohydrazide (3e).** Orange powder. Yield 87%, m.p. = 200 °C. IR spectrum, ν, cm^−1^: 1639 (C=C), 1693 (C=O), 2988 (CH), 3434 (NH). ^31^P NMR (243 MHz, DMSO-*d*_6_) δ 25.06. ^1^H NMR (600 MHz, DMSO-*d*_6_) δ 12.49 (s, 1H, NH), 10.12 (s, 1H, NH), 7.86–7.82 (m, 4H, Ar), 7.80–7.76 (m, 6H, Ar), 7.21–7.18 (m, 1H, Ar), 7.16–7.14 (m, 1H, Ar), 6.92 (d. d, *J* = 12.6 Hz, *J* = 4.2 Hz, 1H, Ar), 3.58 (d, *J* = 14.0 Hz, 2H, CH_2_). ^13^C NMR (151 MHz, DMSO-*d*_6_) δ 162.4 (d, *J* = 4.9 Hz), 162.2, 158.3 (d, *J* = 243.0 Hz), 138.7, 133.5 (d, *J* = 99.4 Hz), 131.6 (d, *J* = 31.8 Hz), 130.6 (d, *J* = 8.6 Hz), 128.3 (d, *J* = 13.6 Hz), 121.1, 117.7 (d, *J* = 24.0 Hz), 113.7 (d, *J* = 24.6 Hz), 113.1 (d, *J* = 8.0 Hz), 107.7 (d, *J* = 12.4 Hz), 36.1 (d, *J* = 65.2 Hz). MS (ESI): *m*/*z* = 422.17 [M + H]^+^; Found: C, 62.56; H, 3.83; N, 9.83; P, 7.18. Anal. calcd. (%) for C_22_H_17_FN_3_O_3_P: C, 62.71; H, 4.07; N, 9.97; P, 7.35.

***N’*-(6-Bromo-2-oxoindolin-3-ylidene)-2-(diphenylphosphoryl)acetohydrazide (3f).** Yellow powder. Yield 81%, m.p. = 195 °C. IR spectrum, ν, cm^−1^: 1617 (C=C), 1692 (C=O), 3054 (CH), 3460 (NH). Due to the impossibility of obtaining a solution of high concentration, ^1^H, ^31^P, and ^13^C NMR spectra were not recorded. MS (ESI): *m*/*z* = 483.94 [M + H]^+^; Found: C, 54.60; H, 3.48; Br, 16.43; N, 8.60; P, 6.29. Anal. calcd. (%) for C_22_H_17_BrN_3_O_3_P: C, 54.79; H, 3.55; Br, 16.57; N, 8.71; P, 6.42.

**2-Chloroethyl (4-(dimethylamino)phenyl)(2-(2-(5-methyl-2-oxoindolin-3-ylidene)hydrazineyl)-2-oxoethyl)phosphinate (5a).** Yellow powder. Yield 90%, m.p. = 195 °C. IR spectrum, ν, cm^−1^: 1629 (C=C), 1686 (C=O), 3081 (CH), 3400 (NH). ^31^P NMR (162 MHz, DMSO-*d*_6_) δ 36.08. ^1^H NMR (500 MHz, DMSO-*d*_6_) δ 12.42 (s, 1H, NH), 11.06 (s, 1H, NH), 7.49 (d. d, *J* = 8.7 Hz, *J* = 11.7 Hz, 2H, Ar), 7.15 (br. d, *J* = 7.7 Hz, 1H, Ar), 7.10 (s, 1H, Ar), 6.79 (d, *J* = 7.9 Hz, 1H, Ar), 6.59 (d. d, *J* = 8.7 Hz, *J* = 2.6 Hz, 2H, Ar), 4.26–4.05 (m, 2H, CH_2_), 3.82–3.79 (m, 2H, CH_2_), 3.68 (d, *J* = 28.4 Hz, 2H, CH_2_), 2.80 (s, 6H, CH_3_), 2.29 (s, 3H, CH_3_). ^13^C NMR (126 MHz, DMSO-*d*_6_) δ 167.8, 162.4, 152.4, 139.9, 133.9, 132.5 (d, *J* = 11.6 Hz), 131.8, 131.4, 121.2, 119.5, 113.9 (d, *J* = 148.6 Hz), 111.0 (d, *J* = 14.0 Hz), 110.6, 64.0 (d, *J* = 5.5 Hz), 56.0, 43.9 (d, *J* = 7.1 Hz), 35.3 (d, *J* = 85.9 Hz), 20.6. MS (ESI): *m*/*z* = 463.14 [M + H]^+^; Found: C, 54.35; H, 5.02; Cl, 7.47; N, 12.01; P, 6.55. Anal. calcd. (%) for C_21_H_24_ClN_4_O_4_P: C, 54.49; H, 5.23; Cl, 7.66; N, 12.10; P, 6.69.

**2-Chloroethyl (4-(dimethylamino)phenyl)(2-(2-(5-methoxy-2-oxoindolin-3-ylidene)hydrazineyl)-2-oxoethyl)phosphinate (5b).** Orange powder. Yield 85%, m.p. = 201 °C. IR spectrum, ν, cm^−1^: 1599 (C=C), 1686 (C=O), 2922 (CH), 3432 (NH). ^31^P NMR (162 MHz, DMSO-*d*_6_) δ 36.02. ^1^H NMR (600 MHz, DMSO-*d*_6_) δ 12.43 (s, 1H, NH), 10.98 (s, 1H, NH), 7.47 (d. d, *J* = 8.9 Hz, *J* = 10.3 Hz, 2H, Ar), 6.93–6.76 (m, 3H, Ar), 6.58 (br. s, 2H, Ar), 4.24–4.11 (m, 2H, CH_2_), 3.85–3.80 (m, 2H, CH_2_), 3.76 (s, 3H, CH_3_), 3.67 (d, *J* = 17.5 Hz, 2H, CH_2_), 2.78 (s, 6H, CH_3_). ^13^C NMR (151 MHz, DMSO-*d*_6_) δ 167.8, 162.4, 152.4, 135.8, 134.0, 132.4 (d, *J* = 11.3 Hz), 120.3, 117.2, 113.9 (d, *J* = 138.6 Hz), 111.6, 110.9 (d, *J* = 14.1 Hz), 106.3, 64.1 (d, *J* = 3.2 Hz), 55.6, 43.9 (d, *J* = 5.7 Hz), 39.1, 35.3 (d, *J* = 83.9 Hz). MS (MALDI): *m*/*z* = 501.0 [M + Na]^+^; Found: C, 52.48; H, 5.00; Cl, 7.23; N, 11.51; P, 6.27. Anal. calcd. (%) for C_21_H_24_ClN_4_O_5_P: C, 52.67; H, 5.05; Cl, 7.40; N, 11.70; P, 6.47.

**2-Chloroethyl (4-(dimethylamino)phenyl)(2-(2-(5-bromo-2-oxoindolin-3-ylidene)hydrazineyl)-2-oxoethyl)phosphinate (5c).** Orange powder. Yield 87%, m.p. = 197 °C. IR spectrum, ν, cm^−1^: 1618 (C=C), 1693 (C=O), 3086 (CH), 3433 (NH). ^31^P NMR (243 MHz, DMSO-*d*_6_) δ 36.84. ^1^H NMR (400 MHz, DMSO-*d*_6_) δ 12.31 (s, 1H, NH), 11.29 (s, 1H, NH), 7.53–7.45 (m, 3H, Ar), 7.41 (s, 1H, Ar), 6.86 (d, *J* = 8.3 Hz, 1H, Ar), 6.58 (d. d, *J* = 8.3 Hz, *J* = 2.2 Hz, 2H, Ar), 4.26–4.06 (m, 2H, CH_2_), 3.82–3.79 (m, 2H, CH_2_), 3.70 (d, *J* = 18.4 Hz, 2H, CH_2_), 2.80 (s, 6H, CH_3_). ^13^C NMR (101 MHz, DMSO-*d*_6_) δ 167.9 (d, *J* = 6.4 Hz), 161.8, 152.4, 141.1, 133.4, 132.5, 132.4 (d, *J* = 10.8 Hz), 123.3, 121.6, 114.1, 113.8 (d, *J* = 148.5 Hz), 112.8, 110.8 (d, *J* = 13.9 Hz), 64.1 (d, *J* = 4.7 Hz), 56.0, 43.9 (d, *J* = 7.4 Hz), 35.5 (d, *J* = 84.4 Hz). MS (ESI): *m*/*z* = 528.97 [M + H]^+^; Found: C, 45.40; H, 3.87; Br, 15.00; Cl, 6.59; N, 10.45; P, 5.67. Anal. calcd. (%) for C_20_H_24_BrClN_4_O_4_P: C, 45.52; H, 4.01; Br, 15.14; Cl, 6.72; N, 10.62; P, 5.87.

**2-Chloroethyl (4-(dimethylamino)phenyl)(2-(2-(5-chloro-2-oxoindolin-3-ylidene)hydrazineyl)-2-oxoethyl)phosphinate (5d).** Orange powder. Yield 80%, m.p. = 198 °C. IR spectrum, ν, cm^−1^: 1621 (C=C), 1693 (C=O), 3082 (CH), 3411 (NH). ^31^P NMR (243 MHz, DMSO-*d*_6_) δ 36.85. ^1^H NMR (400 MHz, DMSO-*d*_6_) δ 12.31 (s, 1H, NH), 11.28 (s, 1H, NH), 7.48 (d. d, *J* = 8.8 Hz, *J* = 11.1 Hz, 2H, Ar), 7.28 (br. d, *J* = 8.4 Hz, 1H, Ar), 7.25 (s, 1H, Ar), 6.90 (d, *J* = 8.3 Hz, 1H, Ar), 6.58 (d. d, *J* = 8.4 Hz, *J* = 2.0 Hz, 2H, Ar), 4.26–4.06 (m, 2H, CH_2_), 3.82–3.79 (m, 2H, CH_2_), 3.70 (d, *J* = 18.6 Hz, 2H, CH_2_), 2.79 (s, 6H, CH_3_). ^13^C NMR (101 MHz, DMSO-*d*_6_) δ 167.9 (d, *J* = 4.4 Hz), 162.0, 152.4, 132.7, 132.5 (d, *J* = 11.7 Hz), 130.7, 126.5, 121.2, 120.5, 114.1, 113.8 (d, *J* = 148.2 Hz), 112.4, 110.9 (d, *J* = 14.0 Hz), 64.1 (d, *J* = 5.2 Hz), 56.0, 43.9 (d, *J* = 7.5 Hz), 35.5 (d, *J* = 84.9 Hz). MS (ESI): *m*/*z* = 483.05 [M + H]^+^; Found: C, 49.52; H, 4.20; Cl, 14.47; N, 11.35; P, 6.22. Anal. calcd. (%) for C_20_H_21_Cl_2_N_4_O_4_P: C, 49.71; H, 4.38; Cl, 14.67; N, 11.59; P, 6.41.

**2-Chloroethyl (4-(dimethylamino)phenyl)(2-(2-(5-fluoro-2-oxoindolin-3-ylidene)hydrazineyl)-2-oxoethyl)phosphinate (5e).** Orange powder. Yield 95%, m.p. = 203 °C. IR spectrum, ν, cm^−1^: 1626 (C=C), 1689 (C=O), 3077 (CH), 3410 (NH). ^31^P NMR (243 MHz, DMSO-*d*_6_) δ 36.89. ^1^H NMR (400 MHz, DMSO-*d*_6_) δ 12.36 (s, 1H, NH), 11.18 (s, 1H, NH), 7.48 (d. d, *J* = 8.9 Hz, *J* = 11.0 Hz, 2H, Ar), 7.20–7.17 (m, 1H, Ar), 7.06 (d, *J* = 7.3 Hz, 1H, Ar), 6.89 (d. d, *J* = 8.4 Hz, *J* = 3.9 Hz, 1H, Ar), 6.58 (br. d, *J* = 6.6 Hz, 2H, Ar), 4.26–4.08 (m, 2H, CH_2_), 3.83–3.80 (m, 2H, CH_2_), 3.69 (d, *J* = 15.6 Hz, 2H, CH_2_), 2.79 (s, 6H, CH_3_). ^13^C NMR (101 MHz, DMSO-*d*_6_) δ 168.0 (d, *J* = 4.3 Hz), 162.4, 158.2 (d, *J* = 238.1 Hz), 152.5, 138.4, 133.2 (d, *J* = 56.0 Hz), 132.6 (d, *J* = 11.5 Hz), 120.8 (d, *J* = 9.2 Hz), 117.7 (d, *J* = 23.6 Hz), 113.7 (d, *J* = 148.4 Hz), 112.0 (d, *J* = 7.4 Hz), 111.0 (d, *J* = 14.2 Hz), 108.0 (d, *J* = 25.6 Hz), 64.2 (d, *J* = 5.0 Hz), 56.1, 44.0 (d, *J* = 6.9 Hz), 35.5 (d, *J* = 84.7 Hz). MS (ESI): *m*/*z* = 467.07 [M + H]^+^; Found: C, 51.22; H, 4.40; Cl, 7.40; N, 11.80; P, 6.40. Anal. calcd. (%) for C_20_H_21_FClN_4_O_4_P: C, 51.46; H, 4.53; Cl, 7.59; N, 12.00; P, 6.63.

**2-Chloroethyl (4-(dimethylamino)phenyl)(2-(2-(6-bromo-2-oxoindolin-3-ylidene)hydrazineyl)-2-oxoethyl)phosphinate (5f).** Orange powder. Yield 93%, m.p. = 193 °C. IR spectrum, ν, cm^−1^: 1615 (C=C), 1694 (C=O), 3092 (CH), 3205 (NH). ^31^P NMR (162 MHz, DMSO-*d*_6_) δ 35.78. ^1^H NMR (400 MHz, DMSO-*d*_6_) δ 12.28 (s, 1H, NH), 11.30 (s, 1H, NH), 7.56–7.52 (m, 1H, Ar), 7.45 (d. d, *J* = 8.7 Hz, *J* = 11.3 Hz, 2H, Ar), 7.06 (s, 1H, Ar), 6.77 (br. d, *J* = 8.7 Hz, 1H, Ar), 6.56 (d. d, *J* = 8.4 Hz, *J* = 2.5 Hz, 2H, Ar), 4.27–4.08 (m, 2H, CH_2_), 3.83–3.80 (m, 2H, CH_2_), 3.67 (m, *J* = 18.7 Hz, 2H, CH_2_), 2.78 (s, 6H, CH_3_). Due to the impossibility of obtaining a solution of high concentration, ^13^C NMR spectra were not recorded. MS (ESI): *m*/*z* = 527.90 [M + H]^+^; Found: C, 45.37; H, 3.82; Br, 14.95; Cl, 6.60; N, 10.50; P, 5.70. Anal. calcd. (%) for C_20_H_24_BrClN_4_O_4_P: C, 45.52; H, 4.01; Br, 15.14; Cl, 6.72; N, 10.62; P, 5.87.

### 3.2. Biological Studies


**Preanalytical stage**


The in vitro experiments were performed using the blood of healthy male donors aged 18–24 years (a total of 52 donors). The study was approved by the Ethics Committee of the Federal State Budgetary Educational Institution of Higher Education at the Bashkir State Medical University of the Ministry of Health of the Russian Federation (No.1 dated 30 January 2024). Informed consent was obtained from all participants before blood sampling. Blood was collected from the cubital vein using a system of vacuum blood collection, the BD Vacutainer^®^ (Becton, Dickinson and Company, Franklin Lakes, NJ, USA). A 3.8% sodium citrate solution in a 9:1 ratio was used as a venous blood stabilizer.


**Thromboelastography**


TEG was performed using TEG 5000 (Haemoscope Corporation, Niles, IL, USA). In the analysis of thromboelastograms, the general tendency of coagulation (R), functional activity of thrombocytes and fibrinogen (MA, Angle), fibrinolytic activity (CLT), and the physical–mechanical properties of formed clots (G) were determined. As a standard activator for TEG, recombinant tissue factor (Innovin^®^, Dade Behring, Germany) was used [42].


**Anticoagulant and Antiplatelet Activities Study**


The study of the effect on platelet aggregation was performed using the Born method [43] using the aggregometer «AT-02» (SPC Medtech, Moscow, Russia). The assessment of antiplatelet activity of the studied compounds and reference preparations was started with the final concentration of 2 × 10^−3^ mol/L. Adenosine diphosphate (ADP; 20 μg/mL) and collagen (5 mg/mL), manufactured by Tehnologia-Standart Company, Russia, were used as inducers of aggregation. The study on the anticoagulant activity was performed by standard recognized clotting tests using an optical two-channel automatic analyzer of blood coagulation, the Solar CGL 2110 (CJSC SOLAR, Minsk, Belarus). The following parameters were studied: activated partial thromboplastin time (APTT), prothrombin time (PT), and fibrinogen concentrations according to the Clauss method. The determination of anticoagulant activity of the studied compounds and reference preparation was performed in a concentration of 5 × 10^−4^ g/mL using the reagents manufactured by Tehnologia-Standart Company (Barnaul, Russia).


**FACs analysis**


Cytofluorimetric analysis was performed on BD FACS Canto II (Becton Dickinson Immunocytometry Systems, Franklin Lakes, NJ, USA) using original software. The expression of P-selectin on the platelet surface was used as a marker of platelet activation. The binding of fluorescently labeled antibodies against CD62 to blood platelets of healthy donors was measured. In order to do this, platelet-rich plasma samples were diluted 100 times with 0.15 M phosphate salt buffer solution (pH 7.0–7.5), and the studied preparations were incubated for 5 min. To activate platelets, ADP was introduced into the samples to reach a final concentration of 20 micrograms/mL and mixed thoroughly. Activation was carried out for 15 min, after which the cells were fixed by adding a 1% formalin solution. After incubation, platelet-rich plasma samples were stained for 20 min at room temperature with mouse anti-CD62 monoclonal antibodies (mAbs) labeled with APC (Alophycocyanin) (Becton Dickinson Immunocytometry Systems, Franklin Lakes, NJ, USA) according to the manufacturer’s recommendations. The same instrument settings were used for all measurements. At least 10,000 events were counted for each sample. The “platelet window” was distinguished by the parameters of direct (FCS) and small-angle (SSC) light scattering in the logarithmic coordinate scale. The number of positive cells (%) was estimated by expression of CD62.

### 3.3. Statistical Analysis

The data were expressed as mean ± SEM and medians and 25 and 75 percentiles. The Shapiro–Wilk test was used to check the normality of actual data distribution. Statistical comparisons were made using a one-way analysis of variance (ANOVA) followed by Dunnett’s Multiple Comparison tests. The two-way repeated measures (mixed model) ANOVA, followed by Bonferroni posttests, was also used to compare the recognition of two objects. A difference with a *p*-value ≤ 0.05 was considered statistically significant. The statistical analysis was performed using GraphPad Prism 5 (GraphPad Software, San Diego, CA, USA). The IC_50_ values were calculated using the online calculator MLA-Quest Graph™ IC_50_ Calculator (AAT Bioquest, Inc., Pleasanton, CA, USA, 14 February 2021). Statistical analysis was performed using the Mann–Whitney test (*p* < 0.05). Tabular and graphical data contain mean values and standard deviation. The results of the study of the anticoagulant and anti-aggregation activities were processed using the statistical package Statistica 10.0 (StatSoft Inc., Tulsa, OK, USA). Analysis of variance was conducted using the Kruskal–Wallis test. A *p*-value of 0.05 was considered statistically significant.

## 4. Conclusions

A series of new isatin hydrazones bearing phosphine oxide or phosphinate groups was synthesized. Quantum chemical calculations, as well as X-ray analysis data, confirmed the predominance of *Z,syn* spatial isomer in solution as in crystals. The results of the study showed that compounds **3a, 3d, 3e, 3f, 5b, and 5c** exhibit anti-aggregatory activity at the level of acetylsalicylic acid and are more effective than acetylsalicylic acid in prolonging the lag period and expression of CD62 in models of intact platelets of healthy donors. In a model of hyperactivity of the hemostasis system under conditions of excess tissue factor, compound **5d** was identified, which was more effective than acetylsalicylic acid in reducing thrombus formation and changing the mechanical characteristics of the clot, indicating a broad antithrombotic potential of derivatives of this series. Our future research in this field will focus on the detailed design of target molecules with improved antiplatelet properties. This will allow researchers to further understand the potential of hybrid isatin compounds containing a phosphorus atom in their structure.

## Data Availability

Samples of all described compounds are available from the author A. Bogdanov.

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
