# Peer review of "Phosphorus-Derived Isatin Hydrazones: Synthesis, Structure, Thromboelastography, Antiplatelet, and Anticoagulation Activity Evaluation"

_ijms, 2025, doi:10.3390/ijms26136147_

Round 1

Reviewer 1 Report

Comments and Suggestions for Authors

<The reviewer deals with the chemistry section mainly>

This manuscript by Samorodov and Bogdanov group described synthesis, structure elucidation, and biological activity evaluation of twelve novel phosphorous-derived isatin hydrazones. 

The one-step synthesis was implemented by a trivial reported condensation method and the structure elucidation is rationally performed by 1H, 13C, and 31P NMRs.  The 31P NMR analysis technique of the obtained target compounds was rationally performed for the E, Z, syn, anti stereoisomers, although the ratio of the major Z, syn isomer is poor to moderate ranging from 1.1 to 1.3.  The X-ray structure seems to be Z, syn isomer based on the comparison with Ref. [33, 34]. 

The speculative work based on the quantum-chemical calculations and the biologically active assays were rationally performed, and the plausible SAR of anti-PAF activity was well-presented.      

On the whole, the reviewer recommends the publication in the journal, after major revisions described as bellow.

<Comments and suggestions>

  1. Abstract & P. 3, line 3 & Scheme 2; Abbreviation of CAPAH should be provided in the first appearance, for readers’ easier comprehension.
  2. Abstract & Conclusion; “Z, syn” may be better than “Zsyn”.
  3. Abstract; In general, “derivatives 3a, 3d, …” should not be recommended, because the abstract description stands itself; The reader con not understand the structure with the compound number. Additionally, “5-chloro phosphinate 5d” should be altered to “5-chloro type phosphinate derivative 5d
  4. 2, line 6 and Figure 1; The contents of text sentence do not completely match with those in Figure 1. For examples, anti-SARS, anti-anxiety Fosazepam, antibacterial, and Kv5.1 inhibitor do not appear in text sentences with relevant citations.  This issue should be arranged.
  5. 2, line ↑4; isatins with → isatins with 1a-f.
  6. 2, line ↑3; The reported synthetic method for the reaction of isatins with hydrazones should be cited.   
  7. 2, line ↑3; have been univocally proven by → were univocally proven by (conventional literature tense)
  8. Schemes 1 & 2; Numberings of the ring-system (1 to 7) should be provided to the starting isatin formula for readers’ easier comprehension.   
  9. Scheme 3; For readers’ easier comprehension, curly double arrows (↔) should be provided to indicate E, Z, syn, anti between two atoms (R, N) and between two single bonds adjacent to C=N double bonds. In addition, hydrogen bonds (···) between O and H should be indicated. 
  10. 3, line ↑7; O…H → O···H
  11. 3, line ↑2; “(Table 1)” should be moved to the end of the sentence (standard style for the literature).
  12. Table 1; The brief discussion for “gas phase conditions” should be provided in the text.
  13. Table 1; The brief discussion for “LUMO-HOMO interaction” should be provided in the text.
  14. 1.2 X-ray study; Brief discussions on E/Z and syn/anti for the obtained crystals should be provided.     
  15. 5, The last sentence; “Next, we will …, and at the end, we will …” is repetitive and not concise. Two “will” may be inappropriate and eccentric.  Refine the sentence.    
Comments on the Quality of English Language

 The English could be improved to more clearly express the research.

Author Response

This manuscript by Samorodov and Bogdanov group described synthesis, structure elucidation, and biological activity evaluation of twelve novel phosphorous-derived isatin hydrazones.

The one-step synthesis was implemented by a trivial reported condensation method and the structure elucidation is rationally performed by 1H, 13C, and 31P NMRs. The 31P NMR analysis technique of the obtained target compounds was rationally performed for the E, Z, syn, anti stereoisomers, although the ratio of the major Z, syn isomer is poor to moderate ranging from 1.1 to 1.3. The X-ray structure seems to be Z, syn isomer based on the comparison with Ref. [33, 34].

The speculative work based on the quantum-chemical calculations and the biologically active assays were rationally performed, and the plausible SAR of anti-PAF activity was well-presented.

On the whole, the reviewer recommends the publication in the journal, after major revisions described as bellow.

Q1. Abstract & P. 3, line 3 & Scheme 2; Abbreviation of CAPAH should be provided in the first appearance, for readers’ easier comprehension.

A1. It was corrected.

Q2. Abstract & Conclusion; “Z,syn” may be better than “Zsyn”.

A2. It was corrected throughout the text as well.

Q3. Abstract; In general, “derivatives 3a, 3d, …” should not be recommended, because the abstract description stands itself; The reader con not understand the structure with the compound number. Additionally, “5-chloro phosphinate 5d” should be altered to “5-chloro type phosphinate derivative 5d

A3. The recommendation was taken into account. The corresponding changes were made to the text.

Q4. 2, line 6 and Figure 1; The contents of text sentence do not completely match with those in Figure 1. For examples, anti-SARS, anti-anxiety Fosazepam, antibacterial, and Kv5.1 inhibitor do not appear in text sentences with relevant citations. This issue should be arranged.

A4. Thank you for your comment. The paragraph below Fig. 1 provides information on the pharmacological importance of phosphine oxides and phosphinates. That is why Fig. 1 shows some examples of chemical structures based on isatin on one side, and phosphorus molecules on the other.

Q5. 2, line ↑4; isatins with → isatins with 1a-f.

A5. It was corrected.

Q6. 2, line ↑3; The reported synthetic method for the reaction of isatins with hydrazones should be cited.

A6. The references used in the work contain more than exhaustive information about the pharmacological attractiveness of isatin. In the cited sources (and references inside) you can see details of obtaining isatin hydrazones.

Q7. 2, line ↑3; have been univocally proven by → were univocally proven by (conventional literature tense)

A7. It was corrected.

Q8. Schemes 1 & 2; Numberings of the ring-system (1 to 7) should be provided to the starting isatin formula for readers’ easier comprehension.

A8. Thank you for the recommendation. In order not to overload Scheme 2, the numbering of atoms in the isatin molecule is given in Scheme 1.

Q9. Scheme 3; For readers’ easier comprehension, curly double arrows (↔) should be provided to indicate E, Z, syn, anti between two atoms (R, N) and between two single bonds adjacent to C=N double bonds. In addition, hydrogen bonds (···) between O and H should be indicated.

A9. Thank you for the recommendation. Hydrogen bonds (···) on Scheme 3 between O and H were indicated. In addition, to demonstrate the presence of isomerism, appropriate changes were made to the product structures in Schemes 1 and 2.

Q10. 3, line ↑7; O…H → O···H

A10. It was corrected.

Q11. 3, line ↑2; “(Table 1)” should be moved to the end of the sentence (standard style for the literature).

A11. It was corrected.

Q12. Table 1; The brief discussion for “gas phase conditions” should be provided in the text.

A12. The following footnote was appended to Table 1: bCalculations performed under gas-phase (vacuum) conditions, neglecting intermolecular interactions and environmental perturbations (e.g., solvation, crystal packing, or external fields).

Q13. Table 1; The brief discussion for “LUMO-HOMO interaction” should be provided in the text.

A13. The following footnote was appended to Table 1: aFrontier orbital gap (between the highest occupied and the lowest unoccupied molecular orbitals).

Q14. 1.2 X-ray study; Brief discussions on E/Z and syn/anti for the obtained crystals should be provided.

A14. Among all possible isomers indicated in Scheme 3, it is the Z,syn form that is realized in crystals of compounds 3e and 5e

Q15. 5, The last sentence; “Next, we will …, and at the end, we will …” is repetitive and not concise. Two “will” may be inappropriate and eccentric. Refine the sentence.

A15. It was corrected.

Reviewer 2 Report

Comments and Suggestions for Authors

Review for ijms-3705073

In this original article entitled “Phosphorus Derived Isatin Hydrazones: Synthesis, Structure, Antiaggregational and Anticoagulation Activity Evaluation”, the authors (Samorodov et al.) reported the synthesis of a series of new phosphorus-containing hydrazones based on aryl-substituted isatins. Furthermore, the authors assessed the thromboelastography (TEG), antiaggregational and anticoagulation potentials of these compounds.

The manuscript is acceptable but requires some minor to major improvements. Hereafter some comments revealed after reviewing its current version.

  • First of all, the authors should reduce the similarity percentage. In fact, the iThenticate report exhibits 47% and 18% from the 1% first source (mdpi.com), which is high.
  • The study is technically sound.
  • It would be better to indicate the thromboelastography even in the manuscript title itself
  • It is recommended to add some limitations of the study, as this would have additional value to the manuscript and enrich its discussion.
  • The manuscript conclusion failed to provide future directions or recommendations.
  • The sentences “Thrombosis and thromboembolic …industrialized countries” and “Disturbances in the hemostasis …critical conditions.” can be supported by the following relevant and recent reference doi: 10.1080/13813455.2025.2504169.
  • English language is overall acceptable just minor checking is required.

Author Response

Reviewer 2

In this original article entitled “Phosphorus Derived Isatin Hydrazones: Synthesis, Structure, Antiaggregational and Anticoagulation Activity Evaluation”, the authors (Samorodov et al.) reported the synthesis of a series of new phosphorus-containing hydrazones based on aryl-substituted isatins. Furthermore, the authors assessed the thromboelastography (TEG), antiaggregational and anticoagulation potentials of these compounds.

The manuscript is acceptable but requires some minor to major improvements. Hereafter some comments revealed after reviewing its current version.

Q1. First of all, the authors should reduce the similarity percentage. In fact, the iThenticate report exhibits 47% and 18% from the 1% first source (mdpi.com), which is high.

A1. Dear editors! Thank you for your comment! Unfortunately, we do not see the report file for originality, but we can definitely guarantee that the compounds are new, the research results were obtained for the first time, all literary sources are cited. At the same time, we still reworked part of the text in the process of answering the questions and comments of the reviewers. Some borrowings are due to the experimental section - devices, methods, methodologies, etc., all this is generally recognized and cannot be changed without distorting the essence of the material.

Q2. The study is technically sound.

A2. Thank you!

Q3. It would be better to indicate the thromboelastography even in the manuscript title itself

A3. Dear Editor! Thank you very much, your comment has been accepted and corrected in the text!

Q4. It is recommended to add some limitations of the study, as this would have additional value to the manuscript and enrich its discussion.

A4. The suggestion has been taken into account. The corresponding phrases have been added. Thank you!

Q5. The manuscript conclusion failed to provide future directions or recommendations.

A5. The suggestion has been taken into account. The corresponding phrases have been added. Thank you!

Q6. The sentences “Thrombosis and thromboembolic …industrialized countries” and “Disturbances in the hemostasis …critical conditions.” can be supported by the following relevant and recent reference doi: 10.1080/13813455.2025.2504169.

A6. Dear Editor! Thank you very much, your comment has been accepted and corrected in the text!

Q7. English language is overall acceptable just minor checking is required.

A7. It was checked. Thank you!

Reviewer 3 Report

Comments and Suggestions for Authors

In this paper, the author synthesized a series of new isatin hydrazones which had the ratio of the two spatial isomers varied from 1 : 1 to 1:3 in most cases out of 12 examples in solution. Quantum-chemical calculations and x-ray analysis were consistent with  the observation that Zsyn form both in gas phase and in solution. Compound 5d exhibitedits antiaggregant properties more effectively than acetylsalicylic acid under the conditions of the tissue factor (TF)-activated thromboelastography (TEG) model. The results suggestted that derivatives had a high potential as a scaffold for the development of effective anticoagulant and antiplatelet agents. The manuscript is well-organized, with a logical flow from introduction to conclusions. The experimental data are robust and statistically rigorous. Overall, I recommend this work for publication in International Journal of Molecular Sciences afetr minor revisions.

Some questions and suggestions are listed as below:

1 In the title, “Antiaggregational and Anticoagulation Activity Evaluation” should be modified as “Antiplatelet and Anticoagulant Activity”. The word “Antiplatelet” is more formal than “Antiaggregational”.

2 In page 5, Section 2.2. Biological Studies, the author described that the studies included experiments to determine antioxidant status, influence on the blood clotting system. and cytotoxic, hemotoxic, and antimicrobial activity. However, i did not see the results of cytotoxic, hemotoxic, and antimicrobial activity.

3 Did the detection of antiplatelet and anticoagulant activity take place in terms of its merged isomeric forms? If so, the activity may not be accurate.

Author Response

Reviewer 3

In this paper, the author synthesized a series of new isatin hydrazones which had the ratio of the two spatial isomers varied from 1 : 1 to 1:3 in most cases out of 12 examples in solution. Quantum-chemical calculations and x-ray analysis were consistent with  the observation that Zsyn form both in gas phase and in solution. Compound 5d exhibitedits antiaggregant properties more effectively than acetylsalicylic acid under the conditions of the tissue factor (TF)-activated thromboelastography (TEG) model. The results suggestted that derivatives had a high potential as a scaffold for the development of effective anticoagulant and antiplatelet agents. The manuscript is well-organized, with a logical flow from introduction to conclusions. The experimental data are robust and statistically rigorous. Overall, I recommend this work for publication in International Journal of Molecular Sciences afetr minor revisions.

Q1. In the title, “Antiaggregational and Anticoagulation Activity Evaluation” should be modified as “Antiplatelet and Anticoagulant Activity”. The word “Antiplatelet” is more formal than “Antiaggregational”.

A1. Dear Editor! Thank you very much, your comment has been accepted and corrected in the text!

Q2. In page 5, Section 2.2. Biological Studies, the author described that the studies included experiments to determine antioxidant status, influence on the blood clotting system and cytotoxic, hemotoxic, and antimicrobial activity. However, i did not see the results of cytotoxic, hemotoxic, and antimicrobial activity.

A2. Dear Editor! Thank you very much, your comment has been accepted and corrected in the text!

Q3. Did the detection of antiplatelet and anticoagulant activity take place in terms of its merged isomeric forms? If so, the activity may not be accurate.

A3. Dear Editor! Indeed, we worked with solutions that contained all the isomers. This was done intentionally, because crystallization of one of the isomers with subsequent dissolution for evaluation of pharmacological properties leads again to transformational changes and formation of isomer complexes. Thank you!
